

**Response patterns of moss to atmospheric nitrogen deposition and nitrogen**
**saturation in an urban-agro-forest transition**
Ouping Deng[a, 1], Yuanyuan Chen[a,1], Jingze Zhao[a], Xi Li[b], Wei Zhou[a], Ting Lan[a], Dinghua Ou[a],
Yanyan Zhang[a], Jiang Liu[a], Ling Luo[c], Yueqiang He[a], Hanqing Yang[d], Rong Huang[a, *]
[a] College of Resources, Sichuan Agricultural University, Chengdu 611130, P. R. China
[b] Ecological Environment Monitoring Station of Sichuan Province, Chengdu 610031, P. R.
China
[c] College of Environmental Sciences, Sichuan Agricultural University, Chengdu 611130, P. R.
China
[d] Chongzhou Meteorological Bureau, Chengdu, 611230, P. R. China
[1] These authors contributed equally to this paper.
*_Correspondence_**:** Rong Huang (14624@sicau.edu.cn)



**Abstract:**

Increasing trends of atmospheric nitrogen (N) deposition resulting from a large
number of anthropogenic emissions of reactive N are dramatically altering the global
biogeochemical cycle of N. Nitrogen uptake by mosses mainly from the atmosphere, making
it a competent bio-indicator for N deposition. However, high uncertainties exist when using
mosses to indicate N deposition, especially in choosing sampling period and sampling
frequency. In this study, atmospheric N deposition and moss N content in the
urban-agro-forest transition, a region with a high N deposition level of 27.46~43.70 kg N
$hm^{-2}$ $yr^{-1}$, were monitored, and the method for atmospheric N deposition monitoring by
mosses was optimized. We found that the optimal sampling frequency is within six months
per time, and the optimal sampling time is autumn (October and November) and summer
(July and August), which gives us a better estimation for atmospheric N deposition than other
scenarios. In addition, the moss N content could better indicate total N deposition than the
deposition of specific N species. This study eventually allowed moss to be used more
effectively and sensibly as an indicator of atmospheric N deposition and helped to improve
the accuracy of the model of quantifying N deposition by using mosses.
**Key words:**
Nitrogen deposition; Moss monitoring; Sampling frequency; Precipitation; Optimal sampling
time



## 1 Introduction

Anthropogenic perturbations have dramatically influenced the nitrogen (N) cycle on the earth's surface (*Vitousek et al., 1997; Galloway et al., 2008*), and much of the excess N originating from agricultural fertilization, animal husbandry, and fossil fuels (including vehicles, energy production, and industry) enters the natural environment (*Meyer et al., 2015*). Atmospheric transport, deposition, and circulation facilitate the conveyance of excessive N to nearby or distant terrestrial and aquatic habitats (*Erisman et al., 2007; Schlesinger, 2009*). As a result, biological and environmental issues, such as water eutrophication, soil acidification, and biodiversity loss, have been reported due to excessive N deposition in some areas (*Clark and Tilman, 2008; Elser et al., 2009; Storkey et al., 2015*). Therefore, it is vital to quantify atmospheric N deposition effectively to provide valuable strategies for N emission mitigation.

Unlike vascular plants, mosses are known to lack a well-developed root system, vascular system and protective cuticle, making them take up water and nutrients primarily from the atmosphere through their surfaces (*Glime, 2007; Keyte et al., 2009; Salemaa et al., 2020*). Hence, mosses have been shown to be suitable indicators of atmospheric deposition, for example, nitrogen (*Pitcairn et al., 2006; Zechmeister et al., 2008; Harmens et al., 2014*) and heavy metals (*Schröder et al., 2010; Harmens et al., 2012*). However, several uncertainties remain in using mosses as a bio-indicator to predict N deposition. First, the sampling frequency (i.e., weeks to years) varied widely in different studies, which largely increased the uncertainty of moss in predicting N deposition. The sampling frequency option will be based on the retention time of mosses for N deposition. It is generally accepted that mosses can preserve the N deposited from the atmosphere for more than one year (*Schröder et al., 2011*). Some studies have also documented that the preservation period of N by mosses is limited (i.e., weeks to months) (*Pavlíková et al., 2016*). Second, the relationship between moss N content and N deposition can vary under different study area conditions. This means that the existing models for N deposition prediction, if used in this study area, may lead to significant uncertainties (*Dong et al., 2017; Wilson et al., 2009*). Third, various forms of N deposition cause distinct responses in mosses. In some N fertilization experiments, mosses were found to prefer ammonium ($NH_4^+$-N) and dissolved organic N (DON) over nitrate ($NO_3^-$-N) as N sources (*Forsum et al., 2006*). Additionally, the natural abundance of N isotopes was used to





find that moss $NO_3^-$-N assimilation was inhibited substantially by the high supply of $NH_4^+$-N
and DON, underscoring the dominance of and preference for atmospheric $NH_4^+$-N in moss N
utilization (*Liu et al., 2013*).
Last, according to current knowledge, N-saturation is defined as the level of pollution
below which there are no significant harmful environmental effects (*UBA, 2005*). N
saturation is widely used in evaluating the impacts of N deposition on ecosystems regarding
excess nutrient N availability, also known as eutrophication (*Burpee and Saros, 2020*). The
absorption of N deposition by moss is limited because N deposition modulates mosses to take
up N by altering their physiological indicators (*Liu et al., 2017; Shi et al., 2017*). Nitrate
reductase is an essential physiological indicator in the N assimilation process of mosses, and
it has been reported that an increase in N deposition leads to a decrease in nitrate reductase,
inhibiting the N uptake and utilization efficiency of mosses (*Arróniz-Crespo et al., 2008;*
*Pearce et al., 2003*). Therefore, N saturation plays a significant role in constraining the
response of moss to N deposition. Above all, it is desirable to improve the moss method for
monitoring atmospheric N deposition from multiple perspectives, especially by improving
sampling parameters. In summary, two questions require resolution to enhance the utilization
of mosses as bio-indicators for predicting N deposition: (i) determining the optimal sampling
period (i.e., sampling frequency and sampling duration) for moss sampling and (ii)
characterizing moss responses and mechanisms to various N deposition forms.
Previous studies have mainly focused on low N deposition ecosystems, such as forests
and grasslands. The urban-agro-forest transition regions include agricultural, urban, rural and
forest areas, which are commonly formed in the process of urbanization and are deeply
influenced by human beings. The patterns and sources of N deposition are more complex
here than in natural ecosystems. However, the method for moss monitoring N deposition is
limited here, and sufficient knowledge is still needed in such high N deposition conditions.
Taking into account the aforementioned limitations, this study conducted a year-long field
experiment to monitor atmospheric N deposition in an urban–agro–forest transition in
Southwest China. The primary objective of this study was to establish a protocol by using
mosses as bio-indicator for the prediction of N deposition. Three aspects were included: (i)
assessing moss responses to atmospheric N deposition, considering variations in sampling



frequency and season; (ii) evaluating the N saturation state of mosses in regions with high N
deposition; and (iii) analyzing moss responses and mechanisms to different N species.

**2   Materials and methods**
**2.1  Study sites**
The field experiment was performed from April 2018 to September 2019 in an
urban-agro-forest transition zone situated in the southwestern Chengdu Plain (Fig. 1). Moss
collection started in October 2018. The climate is subtropical monsoon humid, with a mean
annual temperature, relative humidity, and precipitation of 15.7 °C, 85% and 1103 mm,
respectively. The study encompassed five distinct sites strategically chosen within the
urban-agro-forest transition. These sites represented the four primary land-use types, namely,
agricultural areas (Qiquan, QQ), urban areas (Chongyang, CY), rural areas (Yuantong, YT
and Huaiyuan, HY), and forest areas (Jiguan Mountain, JGM) (Fig. 1). More details about the
study sites are shown in Table S1.

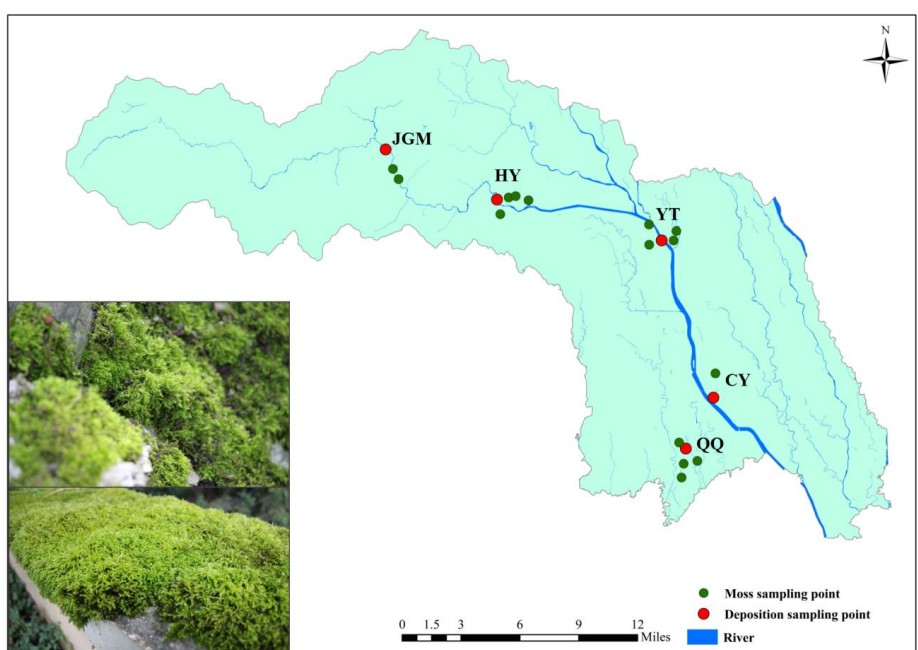


**Figure 1.** Location of the sampling sites. QQ, Qiquan, agricultural areas; CY, Chongyang,
urban areas; YT, Yuantong, rural areas; HY, Huaiyuan, rural areas; JGM, Jiguan Mountain,



forest areas. A field photo of the moss collection is shown in the lower left corner, illustrating
the moss species and sampling substrate.
**2.2 Deposition sampling, analysis, and calculation**
Atmospheric bulk deposition samplers were used to collect N bulk deposition at five
sites, with three parallel samplers at each location to ensure three replicate data. Deposition
samplers were preclean glass cylinders (inner diameter × height of 10.5 cm × 14.5 cm) and
were installed at a height of 1.2 m above the ground with no obstacles and tall buildings
around each site to prevent contamination from surface soil and plants. A stainless-steel net
(pore size, $0.02 \times 0.02 \ m^2$) was used to avoid disturbance of birds, disturbance and crop
stubble contamination. Ultra-pure water was added to each collector, and the depth was kept
at approximately 10 cm (*Wang et al., 2013*). Deposition sampling was conducted at
one-month intervals. The samples were transferred into preclean glass bottles and transported
to the laboratory to determine different forms of N deposition, including dissolved organic N
(DON) and inorganic N ($NH_4^+$-N and $NO_3^-$-N) concentrations, within the same day. Filtered
samples (using 0.45 μm filter membranes) were used for $NH_4^+$-N and $NO_3^-$-N measurements
using an ultraviolet spectrophotometer (UV-1100, Meipuda, China). Unfiltered samples were
collected for total N (TN) measurement through the alkaline potassium peroxydisulfate
oxidation method (APOM). Dissolved organic N (DON) was then calculated using TN
subtracted from the sum of inorganic N (i.e., $NH_4^+$-N and $NO_3^-$-N). It should be noted that
some insoluble N compounds may overestimate the DON contents in this study.
An estimate of bulk deposition in the sampling fluid could be obtained by multiplying
the concentrations by precipitation amounts as follows:

$$F_w = \sum_{i=1}^{n} \frac{C_i \times P_i}{100}$$

(Eq. 1)

where $F_w$ is the flux of N types in monthly deposition, kg N $hm^{-2}$ $mon^{-1}$; $C_i$ is the
concentration of N types in monthly collected samples, mg N $L^{-1}$; $P_i$ is the monthly
precipitation amount, mm; and $i$ represents each month. The precipitation data used in this
study are from the Chongzhou Meteorological Bureau, Sichuan Province, China.




### 2.3 Moss sampling and analysis

The moss materials (*Haplocladium microphyllum* (Hedw.) Broth. subsp. capillatum (Mitt.) Reim.) at all study sites were sampled. This species was chosen based on its larger presence under different growing conditions in this study area, which made the study possible. Moss sampling and preparation were conducted according to guidelines in the ICP Vegetation (ICP Vegetation, 2010), and temporal and spatial synchronization were maintained with deposition sampling. Moss samples were collected every month, which was consistent with collecting N deposition. In this study, 2-5 subsample sites were selected for moss collection within 1 km of the N deposition sampling site (Fig. 1), with at least three replicates of mosses collected from each subsample site. Later, those replicates representing the same deposition sampling site were combined into a representative one. Each subsample was of similar weight and distributed homogenously and as separated as possible within the area, avoiding the collection of concentrated mops within the areas.

All mosses were collected from natural rocks without canopies or overhanging vegetation to avoid the effect of throughfall N compounds. The sampling sites are more than 300 m away from the main roads and at least 100 m away from other roads or houses, free of the direct impact of stagnant water and surface water splashes, traffic, and other artificial pollution sources (human and animal excrement, fertilization, and stamping). The moss samples were stored in polythene zip-lock bags. Dead branches, leaves, and debris attached to the mosses were removed in the lab. Separation of green and brownish parts from mosses for analysis. Only the green part was analyzed, and the brownish part was removed (*Harmens et al., 2014*). After drying the mosses to constant weight in a forced-air oven (at 40°C for 48 h), they were ground to a powder for the moss N content, which was measured by the *Kjeldahl* method after $H_2SO_4$-$H_2O_2$ digestion.

### 2.4 Correlation between moss N content and atmospheric N deposition

The correlation between the moss N content and various atmospheric N deposition under different accumulation time scales (1, 3, 6, 9, and 12 months) was analyzed. This approach enabled the study to discern the appropriate sampling frequency for continuous monitoring of N deposition, revealing that the moss N content in this month exhibited responsiveness to the



cumulative N deposition of preceding months. For example, to analyze the correlation between moss N content in October 2018 and N deposition under the sampling frequency of three months, the value of moss N content should be given as values in October 2018, while the N deposition should be the sum of August, September and October 2018.

Furthermore, correlations between moss N content and various species of N deposition were analyzed in different sampling months, which could obtain the optimal sampling time for moss response to atmospheric N deposition. Note that the time scale of the moss N content is from October 2018 to September 2019, while the N deposition collection period is more than one year, from April 2018 to September 2019, which could enhance the optimality of the sampling frequency for this study.

**2.5  Response model of moss N content to deposition of different N species**

Linear and logarithmic regression analyses of moss N content were fitted to various atmospheric N deposition in SPSS® (version 25.0). Notably, the analysis was carried out at a sampling frequency of one month. The moss N content is the dependent variable, and monthly atmospheric N deposition is the independent variable. The R-squared values derived from observations were instrumental in evaluating the model's optimal fit to the data, thereby aiding in the selection of the most suitable regression approach.

**2.6  Statistical analyses and quality assurance and control (QA/QC)**

Pearson correlation analysis with a two-tailed significance test was used to examine the relationship between moss N content and bulk N deposition, including different sampling times and frequencies. All studies were conducted using SPSS® 25.0 (SPSS Inc., Chicago, USA).

Utmost care was taken to avoid any contamination during the sampling and analytical programme. For the quality assurance (QA) of moss N content measurement, three replicates of each sample were analyzed to provide a stable determination process. Additionally, quality control (QC) was ensured by using certified reference material and laboratory standards for N determination. Additionally, for the determination of the elemental concentrations in the reference material, laboratories followed the same analytical procedure as used for the collected samples.



## 3   Results

### 3.1 Monthly variation in N deposition and moss N content

The range of total N (TN) deposition fluxes in this study was $1.00 \sim 6.44$ kg N $hm^{-2}$ $mon^{-1}$ during the monitoring period from October 2018 to September 2019, which was significantly higher in summer than in other seasons (Fig. S1a, $P < 0.05$). $NH_4^+$-N was the predominant form of N deposition, which ranged from $0.20 \sim 3.89$ kg N $hm^{-2}$ $mon^{-1}$, followed by $NO_3^-$-N $(0.13 \sim 2.33$ kg N $hm^{-2}$ $mon^{-1})$ and DON $(0.00 \sim 1.46$ kg N $hm^{-2}$ $mon^{-1})$. In addition, the different N forms displayed distinct patterns of seasonal variation (Fig. S1). Notably, $NH_4^+$-N, $NO_3^-$-N and DON attained their peak values during the summer and spring seasons.

Mosses in the study area had N contents of $1.51\% \sim 2.96\%$. Notably, the monthly fluctuations in moss samples from the five designated sites displayed a notable similarity. The curve depicting the monthly average variation in moss N contents showed characteristics characterized by a single valley value along with several peaks (Fig. 2a-e). The valley values were commonly observed in the range of January to March. The lowest value was in February (JGM, 1.51%), while the highest was in August (YT, 2.96%). The variation in the N content in moss highly matched the monthly fluctuation patterns of N deposition (all N species) at all study sites (Fig. 2f).



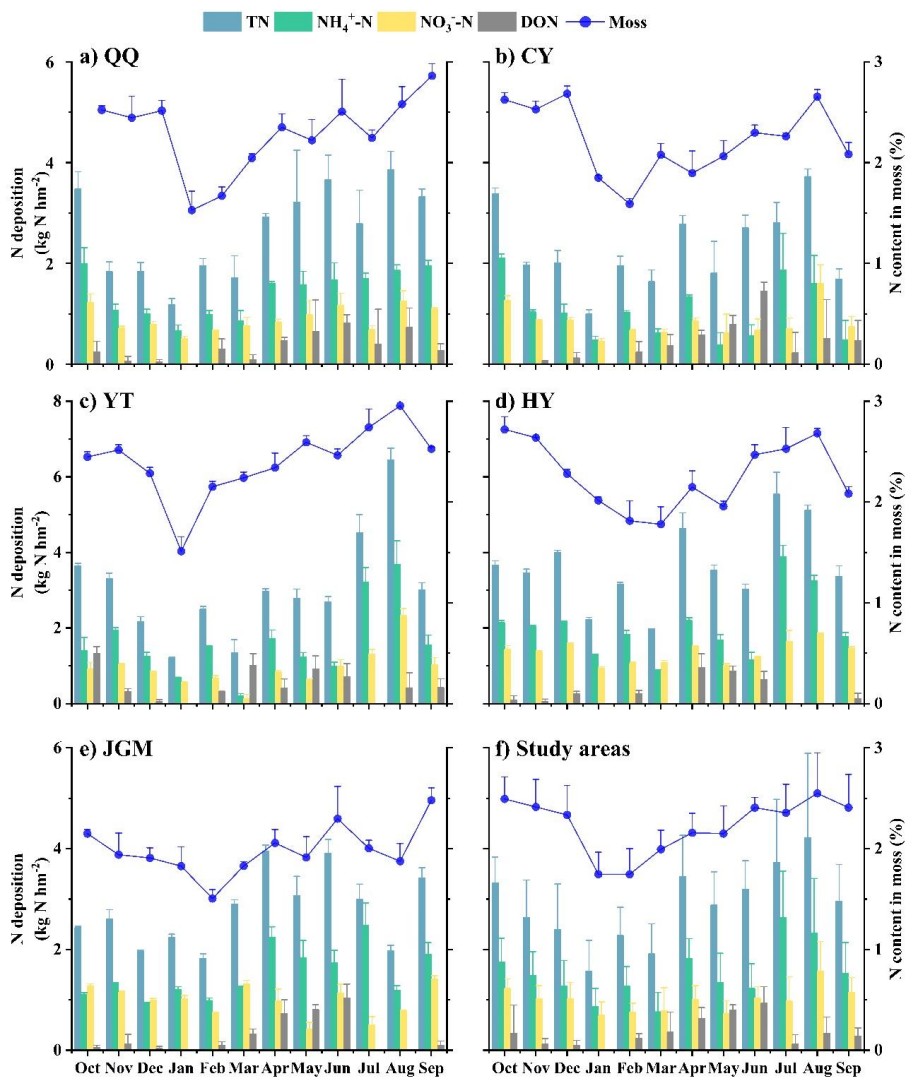

**Figure 2.** Temporal variations in atmospheric N deposition and moss N content at different sites. This figure depicts a year-long (October 2018 - September 2019) overview of N deposition dynamics and moss responses at QQ (a), CY (b), YT (c), HY (d), JGM (e), and Study areas (f), with columns showing deposition data on the left axis and moss N content variations shown as a line on the right axis. Error bars represent the standard deviations of three replicates.

**3.2 Correlation between moss N content and N deposition**

Different N species (TN, $NH_4^+$-N, $NO_3^-$-N, and DON) were used to analyze the



correlation between N deposition and moss N content (Table. 1). The results showed that
when the sampling frequency of mosses was within six months (i.e., every 1, 3, and 6
months), significantly positive correlations ($P < 0.05$) between N species in deposition and
the N content of moss were observed. However, at a sampling frequency of one year (i.e., 12
months), the moss N content and $NO_3^-$-N deposition were found to be negatively correlated
($R$=-0.293, $P < 0.05$).

**Table 1.** Correlation coefficients between moss N content in the current month and N
deposition accumulation in the study area under different sampling frequencies (from one
month per time to one year per time).

| N species | Sampling frequencies | | | | |
|---|---|---|---|---|---|
| | One month | Three months | Six months | Nine months | One year |
| TN | **0.589**** | **0.615**** | **0.370**** | -0.005 | -0.112 |
| $NH_4^+$-N | **0.511**** | **0.532**** | **0.323**** | 0.074 | -0.080 |
| $NO_3^-$-N | **0.517**** | **0.390**** | 0.125 | -0.206 | **-0.293*** |
| DON | 0.114 | **0.460**** | **0.602**** | 0.157 | 0.205 |

**Note:** "**" and "*" indicate $P < 0.01$ and $P < 0.05$, respectively.

Based on the sampling frequency (more than six months per time) that showed a

significant positive correlation, the preferred sampling season was further studied using
correlation analysis (Fig 3). Under the sampling frequency of one month, the moss N content
showed a significant positive correlation with TN-N, $NH_4^+$-N, and $NO_3^-$-N deposition in
winter (January and February), summer (July and August) and autumn (October and
November) ($P < 0.05$). Moreover, DON deposition in spring (March) also showed an exact
correlation with the moss N content. Under the sampling frequency of three months per
sampling event, the correlations between moss N content and N deposition were similar to
those under the sampling frequency of one month per sampling event. Under the sampling
frequency of six months per sampling event, significant positive correlations were observed
only in late autumn and winter, particularly for $NH_4^+$-N.





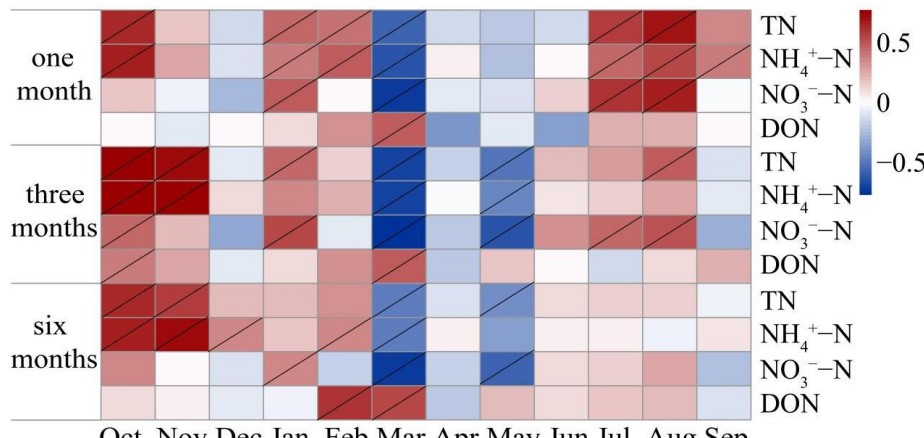

**Figure 3.** Pearson correlation between moss N content in the current month (from left to right) and cumulative N deposition values at different accumulation times covering all sites. The gray slash indicates significance at $P < 0.05$.

### 3.3 Response model and N-saturation state

Both linear and logarithmic models were used to evaluate the response of moss N content to different forms of N deposition (Fig. 4). There were linear and logistic regression relationships between TN, $NH_4^+$-N and $NO_3^-$-N and moss N content. At the same time, there was no relationship between DON and moss N content. The logarithmic models had a high $R^2$ ($P < 0.05$) for TN. However, for $NH_4^+$-N and $NO_3^-$-N, the linear models had high $R^2$ values ($P < 0.05$). Here, the increase in moss N content along the atmospheric N deposition gradient was much faster at a low level than at a high level of atmospheric N input.





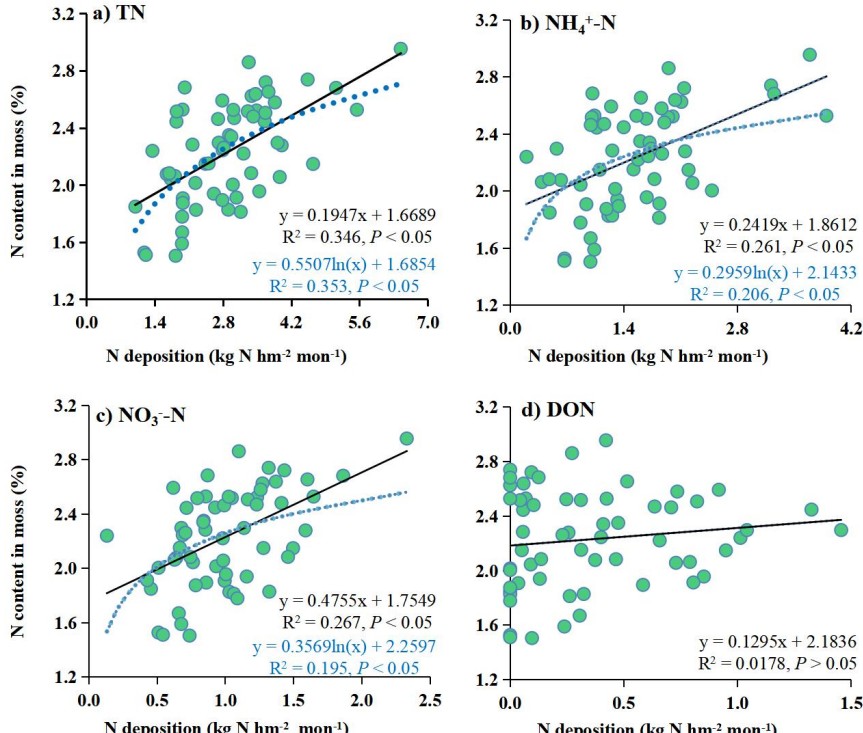

**Figure 4.** Regression relationship between moss N content and bulk N deposition. The nitrogen species considered are TN (a), $NH_4^+$-N (b), $NO_3^-$-N (c), and DON (d), depicted by solid and dashed lines for linear and logarithmic regressions, respectively.

## 4 Discussion

### 4.1 Response pattern to various sampling strategies

Moss N content is a promising indicator in estimating N deposition in the urban–agro–forest transition of this study, owing to the substantial covariation that has been observed (Fig. 2). This viability of mosses in monitoring atmospheric N deposition has been validated through chamber experiments (*Salemaa et al., 2008*). Field sampling in seven European countries has reported that moss N content is correlated with various forms of N deposition (*Harmens et al., 2014*). Due to the physiological characteristics of mosses, especially epilithic mosses, the atmosphere provides a major source of nutrients, not the soil. Therefore, mosses are susceptible to changes in atmospheric N deposition and can also be



used to monitor N deposition. Additionally, mosses can monitor not only atmospheric N
deposition but also atmospheric pollutants, such as heavy metals (*Fernández et al., 2015*).
However, the suitable sampling frequency of mosses remains to be determined.
Theoretically, the higher the sampling frequency is, the more accurate the monitoring of N
deposition. Nevertheless, synergistic monitoring frequencies need to be found due to
financial and other difficulties. In previous studies, mosses were generally believed to retain
N deposition for an extended period (i.e., more than a year), and the relationships between
moss N content and atmospheric N deposition within one-year periods were rarely considered
in these works (*Harmens et al., 2014; Kosonen et al., 2018; Liu et al., 2013*). In this study,
significant covariations between moss N content and N deposition for more than six months
were absent. However, when the sampling frequency of mosses was in the range within six
months (i.e., every 1, 3, and 6 months), significantly positive correlations ($P < 0.05$) between
N species in deposition and the N content of moss were observed. This relation means at least
every 6 months for continuous monitoring of N deposition. The optimal sampling frequency
of moss was explained as the sampling frequency that showed a significant positive
correlation with atmospheric N deposition in this study. This indicates that moss N can only
reflect N deposition in a short period (i.e., less than six months). High atmospheric N
deposition levels in the study region (27.46~43.70 kg N hm$^{-2}$ yr$^{-1}$) can explain this
phenomenon. It has been reported that atmospheric N deposition in Southwest China is
approximately 12.05 kg N hm$^{-2}$ yr$^{-1}$, which is significantly lower than that in this study (*Zhu*
*et al., 2016*). As a result, when the accumulated N deposition exceeds the moss N
sequestration capacity, the responses of moss to atmospheric N deposition may become less
sensitive. Therefore, given the high levels of N deposition observed in this study area, it is
advisable to increase the frequency of moss sampling beyond the current six-month interval
for effective N deposition monitoring. This principle of high-frequency monitoring should
also be extended to regions characterized by substantial N deposition.
The covariation between the moss N content and atmospheric N deposition depends on
the season. For example, significant positive correlations were found between the moss N
content and TN-N, $NH_4^+$-N, and $NO_3^-$-N deposition in autumn (October and November) and
in summer (July and August) (Fig. 3, $P < 0.05$), but these correlations were absent during



winter and autumn. This phenomenon is relevant to the growing season of moss. As
mentioned in several studies, the growth of mosses generally occurs from March to May and
from October to December (*Thöni et al., 2011; Yurukova et al., 2009*). Since mosses undergo
a period of nutrient accumulation during growth (*Faus-Kessler et al., 2001*), they can better
monitor atmospheric N deposition after growth (*Boquete et al., 2011; Thöni et al., 2011*). This
was consistent with the findings of a study that chose to sample mosses between April and
October, which is during the growing season (*Boquete et al., 2011*). The results of this study
also confirm that sampling at this time yields a good correlation between mosses and N
deposition and is one of the appropriate growth intervals for mosses in this study area.

Thus, the optimal sampling season is autumn (October and November) and summer

(July and August) within this area. Moss growth status and regional N deposition level
influence the moss response pattern, subsequently influencing the design of effective
sampling strategies.
**4.2 The response pattern of various species of N**

Significant positive correlations ($P < 0.05$) between various N species in deposition and

the N content of moss were observed when adopting the optimal frequency, i.e., every 1, 3,
and 6 months. The relationship between moss N content and deposition of different N forms
was diverse in this study. Specifically, moss N content correlates better with TN deposition
than other N species. This is consistent with results from several European countries
(*Harmens et al., 2011*).

A comparison among different N species ($NH_4^+$-N, DON, and $NO_3^-$-N) revealed a better

correlation between moss N content and $NH_4^+$-N and DON than $NO_3^-$-N. Notably, at the moss
sampling frequency of six months, the correlation coefficient between DON and moss N
content had the highest R-value ($R=0.602$, $P < 0.01$). This outcome might be attributed to the
adaptability of mosses to their N assimilation processes in response to anthropogenic N
deposition (*Wiedermann et al., 2009*). Research employing $^{15}N$ labeling techniques revealed
that moss displays inducible assimilation of $NO_3^-$-N when $NO_3^-$-N constitutes the sole source
of N, but such assimilation of $NO_3^-$-N becomes negligible in natural environments where the
supply rate of reduced dissolved N ($NH_4^+$-N plus DON) surpasses that of $NO_3^-$-N. The
limited assimilation of $NO_3^-$-N in mosses across different habitats resulted from the inhibition



of nitrate reductase activity, which results from the high supply rate of $NH_4^+$-N plus DON
(*Liu et al., 2012*). In this study, the annual rate of $NH_4^+$-N plus DON (24.21 kg N $hm^{-2}$ $yr^{-1}$)
was 2.03 times greater than that of $NO_3^-$-N (11.91 kg N $hm^{-2}$ $yr^{-1}$). This habitat situation
drives the preference for various N forms for moss uptake. Through $^{15}N$-labeling of $NO_3^-$-N,
$NH_4^+$-N, alanine, and glutamic acids, a previous study found that mosses preferred $NH_4^+$-N
and DON, with deficient uptake of $NO_3^-$-N under different levels of N deposition
(*Wiedermann et al., 2009*). The relatively higher uptake of $NH_4^+$-N than $NO_3^-$-N in moss is
probably due to the high cation-exchange capacity typical for mosses (*Glime, 2007*).
Notably, during autumn (October and November) and in spring (March), there was a
noteworthy and statistically significant positive correlation between the deposition fluxes of
$NH_4^+$-N and DON and the moss N content (Fig. 3, $P < 0.05$). This observed correlation can
be attributed to a main factor. The elevated ambient concentrations of N compounds render
mosses more responsive to atmospheric N deposition. The flux of $NH_4^+$-N deposition was
higher in autumn than in the other seasons (Fig. S1b). This heightened flux in autumn can be
attributed to the peak agricultural activity, including N fertilizer application. It is worth
mentioning that such fertilizer practices lead to ammonia emissions (*Cui et al., 2014*).
Furthermore, the high level of dissolved N nutrients in the topsoil of agricultural land also
facilitates the absorption of N by mosses (*Glime, 2007*). For the same reason, the moss N
content responded better to DON in spring (March). The fluxes of DON were significantly
higher in spring than in autumn and winter in this study (Fig. S1d). It is composed of various
organic compounds, primarily from fossil fuel combustion, and fireworks dominate (*Deng et*
*al., 2018*).
Finally, this study underscores the preference for atmospheric $NH_4^+$-N and DON in moss
N utilization, highlighting the importance of considering the ambient concentration effect on
the response.
**4.3 Relationship between various N forms and the N-saturation state**
Logarithmic models demonstrated a superior fit for the relationship between moss N
content and atmospheric TN deposition (with higher $R^2$, $P < 0.05$) compared to linear models
with the combined dataset encompassing the whole study area (Fig. 4a). This suggests that



the increase in moss N content with increasing atmospheric N deposition is much faster at low levels than at high levels of N deposition.

The utilization of logarithmic models to describe the moss response to N deposition is grounded in the concepts of the "minimum nutrient rate" and the "N-saturation effect". The "minimum nutrient rate" suggests that the growth of crops is influenced by the least available relative concentration of nutrients within the environment. At low N deposition levels, the limitation tends to be N, whereas at high N deposition levels, it may be limited by other nutrients, such as phosphorus. As a result, the rate at which mosses absorb N is influenced by the presence of different limiting nutrients at different N deposition levels, leading to a nonlinear relationship with N (*Vitousek et al., 2010*). Additionally, a distinct "N-saturation effect" has been observed in the relationship between moss N content and N deposition. This phenomenon signifies that there is a point at which the response of mosses to N deposition becomes saturated. When the total N (TN) deposition reaches a state of N saturation, the capacity of mosses to absorb N becomes constrained (*Harmens et al., 2014; Liu et al., 2013; Salemaa et al., 2020*). For instance, when the N deposition level falls below the state of N saturation, mosses display heightened sensitivity to N deposition, leading to significant increases in moss N content. In contrast, when N deposition surpasses the N-saturation state, mosses become less responsive to further N deposition, and the expected increments in moss N content may not materialize. In fact, in such scenarios, the moss N content might even decrease due to growth limitations and physiological disruptions (*Shi et al., 2017*). In summary, the presence of the "minimum nutrient rate" and the "N saturation effect" during deposition influences and restricts the response patterns of mosses.

Notably, the response models constructed using the data from this study indicated that the moss N content exhibited a relatively subdued reaction to TN deposition increases exceeding approximately 4.0 kg N hm$^{-2}$ mon$^{-1}$ (Fig. 4a). This observation suggested that the mosses were approaching the N-saturation state. This phenomenon of N saturation is usually accompanied by a significant decrease in moss abundance and growth, along with the inhibition of photosynthesis and subsequent degradation of chlorophyll (*Britton and Fisher, 2010; Ochoa-Hueso et al., 2013*). These signs could indicate that the threshold of adverse impacts of N on the moss sampled becomes apparent when TN deposition reaches 4.0 kg N



hm$^{-2}$ mon$^{-1}$. The N-saturation state in this study is higher than that in other field studies
conducted in European countries (1.2 and 1.7 kg hm$^{-2}$ mon$^{-1}$, *Harmens et al., 2014, 2011*). It
was also higher than a large number of fluxes on a global scale, such as in Atlantic oak woods
(0.9-1.5 kg hm$^{-2}$ mon$^{-1}$; *Mitchell et al., 2005*) and Yunnan montane forest (1.5 kg hm$^{-2}$ mon$^{-1}$;
*Shi et al., 2017)*. These results could be attributed to the study area being located in a
traditionally high N deposition region in China (*Deng et al., 2018*) because it includes
agricultural, urban, rural and forest areas, which are commonly formed in the process of
urbanization and are deeply influenced by human beings. Therefore, moss species
composition adapted to the elevated N deposition levels in this region. In locations marked by
elevated N pollution, species that are more tolerant tend to thrive over sensitive ones (*Munzi*
*et al., 2019*).
In conclusion, the N-saturation rate exhibited by mosses is significantly influenced by
the atmospheric N deposition background, and this phenomenon displays substantial spatial
variation. Notably, this rate has been determined to be 4.0 kg N hm$^{-2}$ mon$^{-1}$ in the specific
study area under consideration.
Additionally, Fig. 4 shows the relationship between the moss N content and the various
forms of bulk N deposition ($NH_4^+$-N and $NO_3^-$-N). The results showed that linear models
could better fit the moss N content and atmospheric $NH_4^+$-N and $NO_3^-$-N deposition than
logarithmic models (with higher $R^2$, $P < 0.05$) (Fig. 4b, c). This suggests that the increase in
moss N content with increasing atmospheric N deposition is the same at low levels as at high
levels of N deposition. Therefore, the moss N content responds differently to various forms of
N deposition. This provides a new perspective on monitoring N deposition by mosses, which
allows $NH_4^+$-N and $NO_3^-$-N deposition to be observed separately.
**4.4 An optimal guide by using moss to predict atmospheric N deposition**
The following parameters should be noted to improve this technique's accuracy in using
moss to indicate atmospheric nitrogen deposition. First, the optimal sampling frequency and
sampling time are determined. Mosses should be sampled more frequently than every six
months and during autumn (October and November) and summer (July and August) as a
method of monitoring N deposition. Second, the moss N content correlated best with TN
deposition, followed by $NH_4^+$-N, DON and $NO_3^-$-N. Additionally, the application of this



method requires certain preconditions. Understanding the background deposition to determine a more appropriate relationship model and quantify N deposition.

In summary, improving the accuracy of using moss as an indicator for atmospheric nitrogen deposition involves optimizing sampling frequency and timing, recognizing the correlation hierarchy among different nitrogen species, and ensuring that certain preconditions are met for accurate results. Nonetheless, it is important to acknowledge the limitations of this method. First, the method is contingent upon the specific environment where mosses thrive; for instance, it necessitates the collection of epilithic mosses and demands that they be situated in an unshaded area. Second, spatial limitations exist when applying quantitative relationships.

## 5 Conclusion

The moss technique remains a valuable tool for cost-effectively identifying areas at risk of high N deposition, with this study optimizing its parameters. First, the optimal sampling frequency is within six months per time. Second, the optimal sampling periods are autumn and summer, the growing period, allowing for a more accurate estimation of atmospheric N deposition. Third, moss N content exhibited the strongest correlation with TN deposition, indicating its heightened sensitivity to TN deposition. In addition, a new perspective on monitoring N deposition by mosses allows $NH_4^+$-N and $NO_3^-$-N deposition to be observed separately. Enhancing the model's accuracy in quantifying N deposition includes grasping background N deposition values. Considering that some limitations exist, further research is needed on moss response patterns to atmospheric N deposition in various ecosystems across China, particularly those with high N exposure levels.

**Data availability.** Data will be made available on request.

**Author contributions.** OPD and YYC designed the research and collected data. JZZ, YYC and XL wrote the original draft. OPD, RH and JL contributed to review and editing. LL, WZ and TL contributed visualization and validation. DHO, YYZ, YQH and HQY curated the data.



All coauthors were actively involved in extended discussions and the elaboration of the final
design of the manuscript.

**Competing interests.** The authors declare that they have no known competing financial
interests or personal relationships that could have appeared to influence the work reported in
this paper.

**Acknowledgments.** We thank the researchers for field sampling. We appreciate the
meteorological data from the Chongzhou Meteorological Bureau, Sichuan Province, China.

**Financial support.** This research has been supported by the National Natural Science
Foundation of China (grant nos. 42007212 and 42107247), the Sichuan Province Science and
Technology Support Program, China (grant nos. 2022NSFSCO100) and the Natural Science
Foundation of Guizhou Province (Qian- Ke-He-Ji-Chu ZK [2023] Yi ban 474).

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
