# Peer review of "Response patterns of moss to atmospheric nitrogen deposition and nitrogen saturation in an urban-agro-forest transition"

_EGUsphere, 2023_

## Author Comment (AC1)

**Responses to the comments made by Reviewer#1**

**Dear Reviewer:**

Thank you very much for helping us to handle the manuscript entitled "Response patterns of moss to atmospheric nitrogen deposition and nitrogen saturation in an urban-agro-forest transition" (egusphere-2023-2718). I am writing a response to the reviewer's comments. The detailed revisions are highlighted in yellow in the manuscript, and the responses to the comments are listed as follows:

**Q1: Lines 27-28: It is not clear that a better estimation. How to evaluate sampling period? Similar to the lines 28-29, it is not clear about "better indicate"**

**A:** Thank you very much for your comments. We changed "*a better estimation*" to "*a more accurate estimation*" on page 2, Line 27. Besides, we modified the sentence "*In addition, the moss N content could better indicate total N deposition than the deposition of specific N species.*" to "*In addition, the moss N content serves as a more reliable indicator of total N deposition compared to the deposition of specific N species.*" on page 2, Line 28-29.

**Q2: Line 31: Remove "by using mosses"**

**A:** We deleted "*by using mosses*" on page 2, Line 32.

**Q3: Line 40: Atmospheric deposition is not the only way for the anthropogenic N go back to the surface ecosystem. It is better to introduce all the major pathways for anthropogenic N that are input to the earth surface and then highlight the role of atmospheric deposition.**

**A:** Thank you for your comments. We modified the sentences "*Several pathways of anthropogenic N input into earth surface, including deposition, manure, fertilizer and so on (Gu et al., 2015). Atmospheric transport, deposition, and circulation facilitate the conveyance of excess N to nearby or distant terrestrial and aquatic habitats (Erisman et al., 2007; Schlesinger, 2009). Atmospheric N deposition is an important component of the human-accelerated global N cycle and a serious form of*

*atmospheric pollution (Xu et al., 2019), which results in adverse ecological effects, such as water eutrophication, soil acidification, and biodiversity loss, have been reported due to excessive N deposition in some areas (Clark and Tilman, 2008; Elser et al., 2009; Storkey et al., 2015). Atmospheric N deposition has climbed by three-to-five-fold over the course of the 20th century (IPCC 2013). Global N deposition was estimated at 119 Tg N in 2010 (land, 60%; seas, 40%) (Liu et al., 2022).*" to "*Anthropogenic perturbations have dramatically influenced the nitrogen (N) cycle on the earth's surface (Vitousek et al., 1997; Galloway et al., 2008). Several pathways of anthropogenic N input into earth surface, including deposition, manure, fertilizer and so on (Gu et al., 2015). Atmospheric transport, deposition, and circulation facilitate the conveyance of excess N to nearby or distant terrestrial and aquatic habitats (Erisman et al., 2007; Schlesinger, 2009). Atmospheric N deposition is an important component of the human-accelerated global N cycle and a serious form of atmospheric pollution (Xu et al., 2019), which results in adverse ecological effects, such as water eutrophication, soil acidification, and biodiversity loss, have been reported due to excess N deposition in some areas (Clark and Tilman, 2008; Elser et al., 2009; Storkey et al., 2015). Atmospheric N deposition has climbed by three-to-five-fold over the course of the 20th century (IPCC 2013). Global N deposition was estimated at 119 Tg N in 2010 (land, 60%; seas, 40%) (Liu et al., 2022).*" on page 3, Line 37-49.

**Q4: Line 55: What is the retention time of mosses for N deposition? Do you mean the time duration that the N accumulated in mosses?**

**A:** Thanks for your comments. We modified the sentence "*The sampling frequency option will be based on the retention time of mosses for N deposition.*" to "*The sampling frequency option will be based on the time duration that the N deposition accumulated in the mosses.*" on page 3, Line 59-60.

**Q5: Lines 55-56: How about the assimilation of N by moss? Is this amount could be ignored with the N accumulated from deposition?**

**A:** Nitrogen utilization by plants encompasses both nitrogen uptake and assimilation processes. Moss assimilation occurs through the uptake of nitrogen from nitrogen deposition (mainly $NH_4^+$-N, $NO_3^-$-N and DON), which is subsequently assimilated into usable amino acids. It was detailed explained by article titled "*Ammonium first: natural mosses prefer atmospheric ammoniumbut vary utilization of dissolved organic nitrogen depending onhabitat and nitrogen deposition*", authored by Liu et al., in 2013.

**Line 57-58: Why limited? Depend on the moss species?**

**A:** We modified the sentence "*It is generally accepted that mosses can preserve the N deposited from the atmosphere for more than one year (Schröder et al., 2011). Some studies have also documented that the preservation period of N by mosses is limited (i.e., weeks to months) (Pavlíková et al., 2016).*" to "*It is generally accepted that mosses can preserve the N deposited from the atmosphere for more than one year. While some studies have also shown that the preservation period of N by mosses is limited by land use types and moss species, making it possible to maintain N for only a few weeks or months (Schröder et al., 2011; Pavlíková et al., 2016).*" on page 3, Line 60-64.

**Line 59: change "can" to "usually"**

**A:** We modified "*can*" to "*usually*" on page 3, Line 65.

**Line 61: various forms of N from deposition.**

**A:** We modified "*various forms of N deposition*" to "*various forms of N from deposition*" on page 4, Line 68.

**Line 64-67: This sentence is the explanation for the previous mentioned selective accumulation of N by moss. Please change "additionally"**

**A:** We modified "*additionally*" to "*meanwhile*" on page 4, Line 70.

**Line 68: This paragraph still talks about the uncertainties in using mosses as a bio-indicator to predict N deposition. It should be combined with previous paragraph.**

**A:** Thank you very much for your comments. We have combined them into one paragraph on page 4, Line 73-74.

**Figure 1: Major cities and any N emission hotspots should be marked on the map. The wind direction is also needed.**

**A:** Thanks for your comments. We have reddened the names of the two sites with the highest deposition fluxes based on the levels at each atmospheric N deposition monitoring site, as indicated in Figure 1 on page 6, Line 116.

In addition, we want to clarify that the primary focus of this article is to optimize moss monitoring of atmospheric nitrogen deposition as a method, rather than to explore the sources of nitrogen emissions and their spatial variations.

Therefore, we have not listed the main sources of emissions at each point in detail. In addition, due to the constantly changing wind direction during the study period, we did not include wind direction information in the figure as it may not accurately reflect the monitoring time period.

**Line 117: precleaned**

**A:** We midified "*preclean*" to "*precleaned*" on page 6, Line 126.

**Lines 121-122: Did you considered evaporation of prefilled pure water?**

**A:** Thanks for your comments. We designed the experiment considering the evaporation of prefilled pure water. To address this issue, we developed a stainless-steel net that not only prevents disturbance from birds and crop stubble contamination but also helps reduce evaporation.

**L142: Why did you chose "Haplocladium microphyllum (Hedw.) Broth. subsp. capillatum (Mitt.) Reim." as the only moss species? Is it the dominant species?**

**A:** The primary reason for choosing "*Haplocladium microphyllum* (Hedw.) Broth. subsp. capillatum (Mitt.) Reim." As the only moss species is that, during the experimental design phase, we conducted a field study of moss species in the study area. Our findings revealed that this species is the predominant moss species in the study area, and its abundance provides a suitable foundation for our experiment.

**Lines 159-162: Please combine these sentences.**

**A:** We modified the sentences "*The moss samples were stored in polythene zip-lock bags. Dead branches, leaves, and debris attached to the mosses were removed in the lab. Separation of green and brownish parts from mosses for analysis. Only the green part was analyzed, and the brownish part was removed*" to "*The moss samples, stored in polythene zip-lock bags, had dead branches, leaves, and debris removed in the laboratory before separating green and brownish parts for analysis, with only the green part undergoing analysis and the brownish part being discarded.*" on page 8, Line 167-170.

**Line 166: moss total N content?**

**A:** We modified "*the moss N content*" to "*the moss total N content*" on page 8, Line 175.

**L174-179: Why depositions and mosses sampling had different overall time scales, and how does this discrepancy impact the results?**

**A:** Thanks for your comments. The time scale of the moss N content spans from October 2018 to September 2019, while the N deposition collection period extends beyond one year, from April 2018 to September 2019.

From the sampling period perspective, the collection of atmospheric N deposition began earlier than the collection of moss. This sequencing enhances the precision of our experimental results. If the period for collecting atmospheric deposition is shortened to match that of moss collection, moss samples collected in the initial stages would lack accumulated nitrogen deposition data for correlation analysis. For instance, in October 2018, when calculating the correlation between moss nitrogen content and atmospheric nitrogen deposition, data for the preceding 1, 3, and 6 months of nitrogen deposition could not be obtained, rendering the calculation impossible.

**2: It is not clear about the plot f. Average for the 5 sites?**

**A:** Plot f. in Figure 2. represents the average for the 5 sites. We added the sentence "*Additionally, the averages of atmospheric N deposition and moss N content across five sites were shown in Fig. 2f, providing an overview of temporal variations in the study area. It was found that the variation in the N content in moss highly matched the monthly fluctuation patterns of N deposition (all N species) in the study area.*" on page 10, Line 228-231.

**Table 1. It is not clear that what kind of r was used? Pearson or spearman? The size of the dataset should be given for correlation. n=?**

**A:** Pearson correlation analysis with a two-tailed significance test was used in Table 1. The correlations were performed with each result calculated from N deposition samples (n=60) and moss samples (n=60). We also added "*N deposition samples (n=60) and moss samples (n=60) for each correlation.*" on page 12, Line 251-252.

**Figure 3: Each month?**

**A:** The analysis for Figure 3 is explained in the sentences "*Furthermore, correlations between the moss N content and various species of N deposition were analyzed in each sampling months, which could obtain the optimal sampling time for moss response to atmospheric N deposition.*" on page 8, Line 183-185.

**L237: Please confirm whether you intend to express "more than six months per time" or "less than six months per time".**

**A:** We are very sorry for our mistake. We had now changed "*more than six months per time*" to "*less than six months per time.*" on page 12, Line 253.

**Line 256-257: No linear or logarithmic relationship.**

**A:** As shown in Figure 4, "linear or logarithmic relationship" do exist. In our analysis, we considered a relationship to exist when the $P < 0.05$. Please see Figure 4. on page 14, Line 278-281.

**L304-306: You found positive correlations in autumn (October and November) in L304, while absent in autumn in L306. Please double check.**

**A:** We are very sorry for our mistake. We modified "*Autumn*" to "*Spring*" on page 16, Line 323.

**Some minor edits for language**

**Line 40: excess N**

**A:** We modified "*excessive N*" to "*excess N*" on page 3, Line 45.

**Line 54: mosses**

**A:** We modified "*moss*" to "*mosses*" on page 3, Line 59.

**Line 77-78: in limiting the response of mosses**

**A:** We modified "*in constraining the response of moss*" to "*in limiting the response of mosses*" on page 4, Line 83-84.

**Line 122: maintained at approximately 10 cm**

**A:** We modified "*kept at approximately 10 cm*" to "*maintained at approximately 10 cm*" on page 6, Line 130.

**Line 129: dissolved organic nitrogen**

**A:** We modified "*dissolved organic N*" to "*dissolved organic nitrogen*" on page 7, Line 135.

**Line 160: laboratory**

**A:** We modified "*lab*" to "*laboratory*" on page 8, Line 169.

**Line 162: After the mosses were dried**

**A:** We modified "*After drying the mosses*" to "*After the mosses were dried*" on page 8, Line 171.

**Line 169: Change "this month" to "in the given month"**

**A:** We modified "*in this month*" to "*in a given month*" on page 8, Line 178.

**Line 210: change "displayed a notable similarity" to "moss samples from the five designated sites were notably similar"**

**A:** We changed "*displayed a notable similarity*" to "*moss samples from the five designated sites were notably similar*" on page 10, Line 223-224.

**Line 224: Correlations**

**A:** We changed "*Correlation*" to "*Correlations*" on page 11, Line 239.

**Line 319: Response patterns of mosses to various N species**

**A:** We changed "*The response pattern of various species of N*" to "*Response patterns of mosses to various N species*" on page 16, Line 332.

---

## Author Comment (AC2)

**Responses to the comments made by Reviewer#2**

**Dear Reviewer:**

Thank you very much for helping us to handle the manuscript entitled "Response patterns of moss to atmospheric nitrogen deposition and nitrogen saturation in an urban-agro-forest transition" (egusphere-2023-2718). I am writing a response to the reviewer's comments. The detailed revisions are highlighted in yellow in the manuscript, and the responses to the comments are listed as follows:

**Lines 28-29: I am not clear how to compare this? Please revise this sentence. Moss N can be used to indicate total N deposition. Or Moss N is suggested to be used to indicate total N deposition instead of specific N species.**

**A:** Thank you very much for your comments. We modified the sentence "*In addition, the moss N content could better indicate total N deposition than the deposition of specific N species.*" to "*In addition, the moss N content serves as a more reliable indicator of total N deposition compared to the deposition of specific N species.*" on page 2, Line 28-30.

**Line 101: The reason why choosing this urban-agro-forest transition zone should be introduced.**

**A:** The reasons why we chose this urban-agro-forest transition zone are as follows: First, these regions comprise agricultural, urban, rural, and forest areas, which are typically formed during urbanization and are heavily influenced by human activities. Second, nitrogen deposition patterns and sources are more complex in these areas than in natural ecosystems. Finally, moss monitoring methods for nitrogen deposition are limited in such regions, and there is still a need for sufficient knowledge under high nitrogen deposition conditions. Those reasons were stated as "*The urban-agro-forest transition regions include agricultural, urban, rural and forest areas, which are commonly formed in the process of urbanization and are deeply influenced by human beings. The patterns and sources of N deposition are more complex here than in natural ecosystems. However, the method for moss monitoring N*

*deposition is limited here, and sufficient knowledge is still needed in such high N deposition conditions.*" on page 4, Line 91-95.

**1. Different sampling points for QQ, CY, YT, HY, and JGM. For example, 4 sampling points at site QQ but only 2 points at JGM. Why?**

**A:** The number of moss sampling points selected around different atmospheric N deposition sampling sites varied. This variation is due to differences in field conditions at each site, requiring us to conduct on-site surveys to identify locations suitable for moss collection. All mosses were collected from natural rocks without canopies or overhanging vegetation to avoid the effect of throughfall N compounds. The sampling sites are more than 300 m away from the main roads and at least 100 m away from other roads or houses, free of the direct impact of stagnant water and surface water splashes, traffic, and other artificial pollution sources (human and animal excrement, fertilization, and stamping). Therefore, it cannot be guaranteed that the number of moss collection points around each deposition site will be consistent.

**Lines 121-122: Did you record the volume of water in collectors.**

**A:** During the period of water sample collection, we did not continuously monitor the volume of water in the collectors. Instead, we only measured the volume of water in the collectors monthly when collecting water samples for laboratory analysis. These data were then combined with rainfall data provided by the meteorological station to calculate atmospheric nitrogen deposition flux more accurately.

**Section 2.2: How did you sample the moss? What weight or area? Did you consider the period of moss growing?**

**A:** Details about moss sampling and analysis are described in Section 2.3. We have modified the sentences "*In this study, 2-5 subsample sites were selected for moss collection within 1 km of the N deposition sampling site (Fig. 1), with at least three replicates of mosses collected from each subsample site. Later, those replicates representing the same deposition sampling site were combined into a representative one.*" to "*In this study, 2-5 subsample sites were selected for moss collection within 1*

*km of the N deposition sampling site (Fig. 1). Within a 50-meter range (a square of 50×50 m), 5 to 10 samples were collected to combine into a representative one for each subsample site.*" on page 7, Line 157-160.

Besides, each subsample was of similar weight and distributed homogenously and as separated as possible within the area, avoiding the collection of concentrated mops within the areas.

On this issue of whether the period of moss growth is considered, we provide the following explanation. We collected mosses every month. However, we did not attempt to destroy the moss roots as much as possible during the moss collection process and only used the green part. Afterwards, we considered the period of moss growth when we found that significant positive correlations between the moss N content and TN-N, $NH_4^+$-N, and $NO_3^-$-N deposition in winter (January and February), autumn (October and November) and summer (July and August), but these correlations were absent during winter and spring. This phenomenon is relevant to the growing season of moss. The detailed explanations are provided in ***section 4.1***, "*The covariation between the moss N content and atmospheric N deposition depends on the season. For example, significant positive correlations were found between the moss N content and TN-N, $NH_4^+$-N, and $NO_3^-$-N deposition in winter (January and February), summer (July and August) and autumn (October and November) (Fig. 3, P < 0.05), but these correlations were absent during spring. This phenomenon is relevant to the growing season of mosses. As mentioned in several studies, the growth of mosses generally occurs from March to May and from October to December (Thöni et al., 2011; Yurukova et al., 2009). Since mosses undergo a period of nutrient accumulation during growth (Faus-Kessler et al., 2001), they can better monitor atmospheric N deposition after growth (Boquete et al., 2011; Thöni et al., 2011). Thus, the optimal sampling seasons are winter (January and February), summer (July and August) and autumn (October and November) within this area. Moss growth status and regional N deposition level influence the moss response patterns, subsequently influencing the design of effective sampling strategies.*" on page 15-16, Line 319-331.

**Lines 194-196: Details for the certified reference material and laboratory standards are required.**

**A:** Based on your comments, we added the following sentences "*The certified reference materials used in the experiment all conformed to national standards. The standard solutions of NH4+-N, NO3--N and TN complied with GSB 04-2832-2011, GSB 04-1772-2004 and GSB 04-2837-2011 (b). These certified reference materials were stored and utilized correctly.*" on page 9, Line 208-211.

**2 Subplot f is the study areas. What do you mean the study areas? Average of subplots a-e? You can put this one into SI or change it into average N deposition from study sites.**

**A:** Subplot f is the study area, which is the average of five sites, subplots a-e. The detailed description of Subplot f "*Additionally, the averages of atmospheric N deposition and moss N content across the five sites are shown in Fig. 2f, providing an overview of the temporal variations in the study area. It was found that the variation in the N content in moss highly matched the monthly fluctuation patterns of N deposition (all N species) in the study area.*" This information was added to better illustrate the meaning of Fig. 2f, on page 10, Line 228-231.

**Line 231: change "R" to "r", usually we use lowercase letter to show correlation coefficient.**

**A:** We changed "*R*" to "*r*" on page 12, Line 245 and on page 16, Line 342.

**Table 1. I am not clear about the sampling frequencies. In the Section 2.2, the author only mentioned a one-month interval of sampling.**

**A:** Thank you very much for your comments. Our sampling interval, as mentioned in **Section 2.2**, is indeed one month.The correlation analysis was subsequently divided into two steps.

First, by analyzing the relationship between the moss N content of the current month and atmospheric N deposition under different accumulation time scales (1, 3, 6, 9, and

12 months). This approach enabled identification of the appropriate sampling frequency for continuous monitoring of N deposition, revealing that the moss N content in a given month exhibited responsiveness to the cumulative N deposition of preceding months. For example, to analyze the correlation between moss N content in October 2018 and N deposition under the sampling frequency of three months, the value of moss N content should be given as a value in October 2018, while the N deposition should be the sum of August, September and October 2018.

After analyzing the maximum cumulative N deposition time scale that mosses can respond to within six months, we can refine the moss monitoring method for N deposition, controlling the frequency of moss collection to less than six months per time. This methodology is described in detail in **Section 2.4**, "*The correlation between the moss total N content and various atmospheric N deposition under different accumulation time scales (1, 3, 6, 9, and 12 months) was analyzed. This approach enabled the study to discern the appropriate sampling frequency for continuous monitoring of N deposition, revealing that the moss N content in a given month was sensitive to the cumulative N deposition in the preceding months. For example, to analyze the correlation between the moss N content in October 2018 and N deposition under the sampling frequency of three months, the value of moss N content should be given as a value in October 2018, while the N deposition should be the sum of August, September and October 2018.*", on page 8, Line 175-182.

Afterwards, based on the optimal sampling frequency, the correlations between moss N content and various species of N deposition were analyzed in each sampling months, which could obtain the optimal sampling time for moss response to atmospheric N deposition. This methodology is elaborated in detail in **Section 2.4**, "*Furthermore, correlations between the moss N content and various species of N deposition were analyzed in each sampling months, which could obtain the optimal sampling time for moss response to atmospheric N deposition. Note that the time scale of the moss N contentwas from October 2018 to September 2019, while the N deposition collection period was more than one year, from April 2018 to September*

*2019, which could enhance the optimality of the sampling frequency for this study.*",
on page 8, Line 183-188.

**2 and 3 are all correlations, but presented in different ways. Can you make one figure to show both correlations with similar ways?**

**A:** Thank you very much for your comments. Figures 2 and 3 represent different correlations and meanings.

Figure 2 is described in detail in "***Section 3.1 Monthly variation in N deposition and moss N content***" on pages 9-10. Fig. 2a-e shows the monthly N deposition flux and moss N content at the five sampling sites. Besides, the averages of atmospheric N deposition and moss N content across the five sites are shown in Fig. 2f, providing an overview of the temporal variations in the study area. In this section, the study did not actually calculate the correlation between them.

Figure 3 is described in detail in "***Section 3.2 Correlations between moss N content and N deposition***" on page 12-13, Line 253-264. Figure 3 was based on the optimal sampling frequency, which was calculated in Table 1. The correlations between the moss N content and the various N deposition species were analyzed in each sampling month to determine the optimal sampling time for determining the response of the moss to atmospheric N deposition. This methodology is elaborated in detail in ***Section 2.4 Correlation between moss N content and atmospheric N deposition***.

Above all, we cannot make one figure to show both correlations in similar ways.

**In section 3.3. Both linear and logarithmic regressions were conducted. Here, only one regression is enough. Please selected a better regression with higher R2 Here, only TN follows a linear regression, whereas NH4+-N and NO3--N follow a non-linear regression.**

**A:** In this study, both linear and logarithmic models were used to evaluate the response of moss N content to different forms of N deposition. By displaying both regressions, we aim to highlight the results of this section. The results showed that the logarithmic models had a high $R^2$ ($P < 0.05$) for TN. However, for $NH_4^+$-N and $NO_3^-$-N, the linear models had high $R^2$ values ($P < 0.05$). Based on the results, the relationships between the moss total N content and various N forms, and the N-saturation state, were discussed in ***section 4.3***. Therefore, we consider that both regressions should be conducted simultaneously and are presented in Figure 4.

**In the section of Introduction, N deposition per year should be given.**

**A:** Thank you very much for your comments. Based on your comments, we supplemented "*Atmospheric N deposition has climbed by three-to-fivefold over the course of the 20th century (IPCC 2013). Global N deposition was estimated at 119 Tg N in 2010 (land, 60%; seas, 40%) (Liu et al., 2022).*" in the section of Introduction, on page 3, Line 46-49, to better illustrate the importance of atmospheric N deposition in the global N cycle and the need for its monitoring. Besides, the corresponding references were added to the reference lists on page 24, Line 562-563 and Line 571-574.

*(1)  IPCC. Climate Change 2013-The Physical Science Basis. New York: Cambridge University Press; 2013.*

*(2)  Liu, L., Xu, W., Lu, X., Zhong, B., Guo, Y., Lu, X., Zhao, Y., He, W., Wang, S., Zhang, X., Liu, X., Vitousek, P.: Exploring global changes in agricultural ammonia emissions and their contribution to nitrogen deposition since 1980. Proc. Natl. Acad. Sci., 119, https://doi.org/10.1073/pnas.2121998119, 2022.*

---

## Author Comment (AC3)

**Responses to the comments made by Reviewer**

**Dear Reviewer:**

Thank you very much for helping us to handle the manuscript entitled "Response patterns of moss to atmospheric nitrogen deposition and nitrogen saturation in an urban-agro-forest transition" (egusphere-2023-2718). I am writing a response to the reviewer's comments. The detailed revisions are highlighted in yellow in the manuscript, and the responses to the comments are listed as follows:

**1. In the title part, response pattern, nitrogen saturation, and ubran-agro-forest transition is hard to be understood.**

**A:** First, moss responses to atmospheric N deposition are influenced by factors such as moss sampling time, frequency, and various N deposition forms. Therefore, this study explored these factors that influence moss response, aiming to understand the pattern of moss response to nitrogen deposition and to refine moss monitoring of atmospheric nitrogen deposition. Second, nitrogen saturation represents a phenomenon where mosses become less responsive to further nitrogen deposition, and the expected increases in moss nitrogen content may not occur. In fact, in such scenarios, moss nitrogen content might even decrease due to growth limitations and physiological disruptions. Finally, "ubran-agro-forest transition" refers to the transition zone in this study area, which transitions from urban to agricultural areas and then to forests.

**2. In the introduction part, what is the N saturation state of mosses? I think in the present study, the data can not support the N saturation state of mosses.**

**A:** As mentioned in the introduction part, N-saturation is defined as the level of pollution below which there are no significant harmful environmental effects (*UBA, 2005*). The absorption of N deposition by mosses is limited because N deposition modulates mosses to take up N by altering their physiological indicators (*Liu et al., 2017; Shi et al., 2017*). Thus, there exists a N saturation state for mosses, in which the

rate of increase in moss N content with atmospheric N deposition slows down and may instead show a decreasing trend.

As detailed in "*section 4.3 Relationships between various N forms and the N-saturation state*", the N-saturation state was discussed. In this study, we found that the increase in moss total N content with increasing atmospheric total N deposition was much faster at low levels than at high levels of N deposition. This phenomenon signifies that there is a point at which the response of mosses to N deposition becomes saturated. This also indicates that the moss response to N deposition is indeed influenced by N saturation.

Later, we constructed response models using the data from this study, which indicated that the moss N content exhibited a relatively subdued reaction to TN deposition increases exceeding approximately 4.0 kg N $hm^{-2}$ $mon^{-1}$. This observation suggested that the mosses were approaching the N-saturation state.

Although the data from this study didn't provide an accurate value for the N saturation state of mosses, the phenomenon of N saturation states constraining the response of mosses to N deposition was observed in this study and approximate values for the saturation value of N deposition could be suggested.

**3. In the method part, it is better to give the land use types to show the transition of urbans-agro-forest in the fig. 1.**

**A:** Thank you very much for your comments. We have taken your suggestion into account and redrawn Fig. 1. on page 6, Line 115. In the new version, the land use types are presented. At the same time, we have added the source of the land use data: "*The land-use data (2016) used here were provided by the Center of Land Acquisition and Consolidation in Sichuan Province.*" to the description of the figure on page 6, Line 121-122.

**4. Lines 121-123, why add water to the collector? This will affect quantify precipitation. Besides, the rain samples were collected at one-month interval, which will cause nitrogen loss and the variation of nitrogen forms as cause by microbial activities.**

**A:** Thank you very much for your comments. First, we added water when placing the collectors to collect partial dry deposition through the wet surface method. Second, the volume of the device obtained by measurement, however, was only used to support calculations when accurate precipitation could not be obtained. The actual precipitation data used in this study were provided by the Chongzhou Meteorological Bureau, Sichuan Province, China. Therefore, adding water to the collectors should not affect the quantity of precipitation.

Third, during the preliminary experiments, we also observed the issue raised by the reviewer regarding nitrogen loss. Subsequently, we increased the maximum capacity of our collectors to minimize N loss due to sample overflow. Finally, during the summer, we added 1 mL of 2 mol/L copper sulfate solution to the collectors to prevent the growth of bacteria and algae. However, due to the lower temperatures in winter, we disregard microbial activity. Additionally, we used a stainless-steel net to minimize disturbance from birds and contamination from crop stubble, aiming to reduce microbial proliferation as much as possible. We added the sentences "*During the summer, 1 mL of 2 mol/L copper sulfate solution was added to the collectors to prevent the growth of bacteria and algae.*" on page 6, Line 131-132. Certainly, we will also refine our experiments to address the variation in N forms caused by microbial activities.

**5. Line 133, this study only measured bulk N deposition, will dry N deposition affect moss N content?**

**A:** Conventional monitoring methods for atmospheric N deposition generally include dry deposition, wet deposition, and bulk deposition. The available measurements in China thus far have focused primarily focus on wet or bulk deposition. Although dry

deposition is important, it is more challenging and often cost-prohibitive to measure. Among these methods, the bulk deposition monitoring method utilizes standardized glass deposition collectors to collect bulk deposition, collects liquid from the collectors at regular intervals, and then determines the concentration of N compounds to estimate atmospheric N deposition. This method partially compensates for the drawbacks of dry deposition and is widely applied.

Considering that the study area includes both outdoor and backward facilities, we chose the bulk deposition monitoring method to monitor N deposition in the region. Bulk deposition, which encompasses wet deposition plus a fraction of dry deposition, has recently been incorporated into a 43-site monitoring network across China. On average, bulk and dry N deposition rates were equally significant (50% each) (*Xu et al., 2015*). Additionally, one study reported that trends in bulk deposition may provide a useful guide for increasing in total N deposition rates (*Song et al., 2017*). Therefore, we believe that measuring bulk N deposition is also meaningful.

Above all, the current experimental results cannot accurately determine whether N dry deposition affects moss N content or quantify its impact. We can only quantify the relationship between bulk N deposition and moss N content.

The following are the relevant references cited in this response:

(1)  Xu, W., Luo, X., Pan, Y., et al.: Quantifying atmospheric nitrogen deposition through a nationwide monitoring network across China. Atmos. Chem. Phys. 15, 12345-12360, https://doi.org/10.5194/acp-15-12345-2015, 2015.

(2)  Song, L., Kuang, F., Skiba, U., et al.: Bulk deposition of organic and inorganic nitrogen in southwest China from 2008 to 2013, Environ. Pollut., 227, 157-166, https://doi.org/10.1016/j.envpol.2017.04.031, 2017.

**6. Lines 154-155, moss usually were found in the places with trees. Is it difficult to find mosses living under this condition?**

**A:** Thanks for your comments. Finding mosses that meet the conditions mentioned in the paper "from natural rocks without canopies or overhanging vegetation" is indeed challenging. During site selection, we conducted rigorous field investigations primarily to mitigate sources of N for mosses other than atmospheric nitrogen deposition. These sources include leaf litter deposition and soil nitrogen, which could influence the results.

**7. In the results part, as showed in figure 4, the R2 between N deposition and nitrogen content in moss is low, so how is the accuracy for using moss as the bio-monitor of atmospheric N deposition?**

**A:** In our analysis, we considered the correlation between N deposition and moss N content to be accurate and valid when it met a significance level of $P < 0.05$. The reasons for the low $R^2$ in our analysis are as follows. First, our study covered a wide range. Five distinct sites were strategically chosen within the urban-agro-forest transition. These sites represented the four primary land-use types, each representing one of the four primary land-use types. Second, the time scale of our data spans one year, resulting in a large volume of data. Above all, these factors collectively contributed to the lower $R^2$ of the regression equation.

**8. In the conclusion part, if the moss can only be sampled in autumn and summer, how can it be used for as bio-monitor of N deposition in a year?**

**A:** We are very sorry for our negligence. With your reminder, we referred to Figure 3 and checked the results section "***3.2 Correlations between moss N content and N deposition***". We identified our oversight and made the following modifications.

(1) We modified "*the optimal sampling time is autumn (October and November) and summer (July and August)*" to "*the optimal sampling times are winter (January*

*and February), autumn (October and November) and summer (July and August)"*
on page 2, Line 25-26.

(2)  We modified "*in autumn (October and November) and in summer (July and August)*" to "*winter (January and February), summer (July and August) and autumn (October and November)*" on page 15, Line 321-322.

(3)  We modified "*autumn (October and November) and summer (July and August) within this area*" to "*winter (January and February), summer (July and August) and autumn (October and November)*" on page 16, Line 328-329.

(4)  We modified "*Mosses should be sampled more frequently than every six months and during autumn (October and November) and summer (July and August) as a method of monitoring N deposition.*" to "*Mosses should be sampled more frequently than every six months and during winter (January and February), autumn (October and November) and summer (July and August) as a method of monitoring N deposition.*" on page 19, Line 435-437.

(5)  We modified "*Second, the optimal sampling periods are autumn and summer, the growing period, allowing for a more accurate estimation of atmospheric N deposition.*" to "*Second, the optimal sampling periods were winter, summer and autumn, allowing for a more accurate estimation of atmospheric N deposition.*" on page 20, Line 454-455.

Regarding this issue, we made descriptions as following:

First, sampling conducted during winter (January and February), summer (July and August), and autumn (October and November) was deemed the optimal sampling time. However, this does not render samples collected during other months meaningless; they can still offer qualitative or semi-quantitative insights for monitoring atmospheric N deposition.

Additionally, this study optimized several parameters of moss monitoring for nitrogen deposition. Among them, the optimal sampling time was based on the optimal sampling frequency, which represents the longest cumulative N deposition time scale

to which mosses collected in the current month can respond to. This study revealed that the optimal sampling frequency was within six months per time. When using a bio-monitor for nitrogen deposition, integrating the optimal sampling frequency with the optimal sampling time can not only enhance the accuracy of this method, but also can extend the monitoring time scale beyond just the optimal sampling period to encompass the entire year. We obtained the monthly nitrogen deposition flux by inputting the moss nitrogen content collected in each month into the regression equation (Fig. 4). Then, based on the optimal sampling frequency, we multiply it by the total number of months. By repeating this process, we can derive the annual deposition flux.